# The Abbott PanBio WHO emergency use listed, rapid, antigen-detecting point-of-care diagnostic test for *SARS-CoV-2*—Evaluation of the accuracy and ease-of-use

Lisa J. Krüger[1], Mary Gaeddert[1], Frank Tobian[1], Federica Lainati[1], Claudius Gottschalk[1], Julian A. F. Klein[1], Paul Schnitzler[2], Hans-Georg Kräusslich[2], Olga Nikolai[3], Andreas K. Lindner[3], Frank P. Mockenhaupt[3], Joachim Seybold[4], Victor M. Corman[5,6], Christian Drosten[5,6], Nira R. Pollock[7], Britta Knorr[8], Andreas Welker[8], Margaretha de Vos[9], Jilian A. Sacks[9], Claudia M. Denkinger[1,10]*, for the study team[¶]

1 Division of Clinical Tropical Medicine, Heidelberg University Hospital, Heidelberg, Germany, 2 Virology, Heidelberg University Hospital, Heidelberg, Germany, 3 Charité - Universitätsmedizin Berlin, Corporate Member of Freie Universität Berlin, Humboldt-Universität zu Berlin, and Berlin Institute of Health, Institute of Tropical Medicine and International Health, Berlin, Germany, 4 Charité - Universitätsmedizin Berlin, Corporate Member of Freie Universität Berlin, Humboldt-Universität zu Berlin, and Berlin Institute of Health, Medical Directorate, Berlin, Germany, 5 Charité - Universitätsmedizin Berlin, Corporate Member of Freie Universität Berlin, Humboldt-Universität zu Berlin, and Berlin Institute of Health, Institute of Virology, Berlin, Germany, 6 German Centre for Infection Research (DZIF), Partner Site Charité, Berlin, Germany, 7 Department of Laboratory Medicine, Boston Children's Hospital, Boston, Massachusetts, United States of America, 8 Department of Public Health Rhein Neckar Region, Heidelberg, Germany, 9 Foundation for Innovative New Diagnostics, Geneva, Switzerland, 10 German Centre for Infection Research (DZIF), Partner Site Heidelberg University Hospital, Heidelberg, Germany

¶ Membership of the study team are detailed in S1 File.
* Claudia.denkinger@uni-heidelberg.de

**Data Availability Statement:** All relevant data are available at https://doi.org/10.11588/data/FSPQL4.

## Abstract

### Objectives

Diagnostics are essential for controlling the pandemic. Identifying a reliable and fast diagnostic device is needed for effective testing. We assessed performance and ease-of-use of the Abbott PanBio antigen-detecting rapid diagnostic test (Ag-RDT).

### Methods

This prospective, multi-centre diagnostic accuracy study enrolled at two sites in Germany. Following routine testing with reverse-transcriptase polymerase chain reaction (RT-PCR), a second study-exclusive swab was performed for Ag-RDT testing. Routine swabs were nasopharyngeal (NP) or combined NP/oropharyngeal (OP) whereas the study-exclusive swabs were NP. To evaluate performance, sensitivity and specificity were assessed overall and in predefined sub-analyses accordingly to cycle-threshold values, days after symptom onset, disease severity and study site. Additionally, an ease-of-use assessment (EoU) and System Usability Scale (SUS) were performed.

**Funding:** The study was supported by Heidelberg and Charité University Hospital internal funds awarded to CD, as well as a grant of the Ministry of Science, Research and the Arts of Baden-Württemberg, Germany awarded to CD. Foundation of Innovative New Diagnostics (FIND) reports grants from UK Department of International Development (DFID, recently replaced by FCMO - UK Foreign, Commonwealth & Development Office) (Grant No. 300341-102) awarded to JS, grants from World Health Organization (WHO) awarded to JS, and grants from Unitaid (No. 2019-32-FIND MDR) awarded to JS to conduct of the study.

**Competing interests:** The authors have declared that no competing interests exist.

## Results

1108 participants were enrolled between Sept 28 and Oct 30, 2020. Of these, 106 (9.6%) were PCR-positive. The Abbott PanBio detected 92/106 PCR-positive participants with a sensitivity of 86.8% (95% CI: 79.0% - 92.0%) and a specificity of 99.9% (95% CI: 99.4%-100%). The sub-analyses indicated that sensitivity was 95.8% in Ct-values <25 and within the first seven days from symptom onset. The test was characterized as easy to use (SUS: 86/100) and considered suitable for point-of-care settings.

## Conclusion

The Abbott PanBio Ag-RDT performs well for *SARS-CoV-2* testing in this large manufacturer independent study, confirming its WHO recommendation for Emergency Use in settings with limited resources.

## Introduction

Diagnostics are a cornerstone of pandemic control. In March 2020, the World Health Organisation (WHO) emphasized the importance of access to testing for an effective control of *SARS-CoV-2* infections [1]. While reverse-transcriptase polymerase chain reaction (RT-PCR) remains the gold standard among diagnostic tests for *SARS-CoV-2*, access may be limited due to shortages of instruments, supplies and experienced operators, particularly in resource-limited settings. Antigen-detecting tests (Ag-RDTs) offer an alternative to RT-PCR and have been recommended by the WHO for appropriate settings where nucleic acid amplification technology (NAAT) testing is limited or where prolonged turnaround times slow down clinical testing if they achieve at least ≥80% sensitivity and ≥97% specificity compared to a NAAT reference assay [2].

The Foundation of New Innovative Diagnostics (FIND) has identified >80 antigen tests in the pipeline for *SARS-CoV-2* [3]. To this date, the WHO has recommended two Ag-RDTs on the Emergency Use Listing based on data from manufacturers as well as one independently-conducted accuracy study [4]. These two Ag-RDTs are SD Biosensor STANDARD Q (recommended on Sept 22, 2020) followed by the Abbott PanBio Ag-RDT (Oct 2,2020) [5, 6]. This manufacturer-independent study follows a WHO-approved protocol to complement the data provided to the WHO by the manufacturer.

Several other studies evaluating the Abbott PanBio Ag-RDT have been published to date. Of three large studies only one single-centre study prospectively enrolled participants [7]. The other two selected stored samples with the representativeness of the selection being unclear in one study [8, 9]. One study with a limited sample size (N = 255) concluded that sensitivity was the highest within the first week after symptom onset [10] supported by two additional studies reporting sensitivity to be the highest in participants presenting with a low Ct-value and a respectively high viral load, often observed at the onset of symptoms [11, 12]. The sensitivity was determined to be in the range of 60%-92% in these six studies along with a very high specificity of 98.9% and above [7–12].

The prospective multi-centre clinical accuracy study reported here, represents the largest manufacturer-independent dataset for the point-of-care (POC) performance of the Abbott PanBio Ag-RDT of date and also represents to our knowledge the only comprehensive ease-of-use assessment (EoU) of the Abbott PanBio Ag-RDT.

## Material and methods

### Ethic statement

The study protocol was approved in March 2020 by the ethical review committee at the Heidelberg University Hospital for the two study sites Heidelberg and Berlin in Germany (Registration number S-180/2020).

### Clinical diagnostic accuracy study

The *standards for reporting diagnostic accuracy studies* (STARD) were followed for this study.

The test evaluated in this accuracy study is the PanBio™ COVID-19 Ag Rapid Test Device (Abbott Rapid Diagnostics, Jena, Germany; henceforth called PanBio) [13].

The test uses the lateral flow assay principle in a cassette format design for the detection of viral antigens. The test kits include proprietary swabs for sample collection. As indicated in the instruction for use (IFU), five drops of the extracted specimen in the provided buffer solution are applied to the test device. Colloidal gold conjugated antibodies on the membrane strip react with viral antigens generating a colour change in the device window, which can be interpreted with the naked eye. The results should be interpreted within 15 to 20 minutes of incubation and are considered invalid if interpreted after this timeframe. The manufacturer's IFU were followed during sampling and testing procedures.

**Study design and participants.**   The enrolment of participants was conducted at two sites in Germany; in Heidelberg at a drive-in testing site; in Berlin at a clinical ambulatory testing facility. Inclusion criteria for the participation were age ≥18 years and classification as being at risk for a *SARS-CoV-2* infection by the local health department based on contact with a confirmed *SARS-CoV-2* case or symptoms suggestive for infection. Individuals with a prior positive RT-PCR test for *SARS-CoV-2* or those who could not give written informed consent due to limited command of English or German were excluded. The study protocol is available upon request.

**Study procedures.**   Individuals meeting the inclusion criteria were invited to participate in the study. After providing written informed consent, participants first underwent the routine swab for RT-PCR testing directly followed by the study-exclusive swab for Ag-RDT testing, performed by the trained study team. Routine sampling for RT-PCR testing was performed with a nasopharyngeal (NP) swab in Heidelberg and a combination of a NP and oropharyngeal (OP) swab in Berlin as per institutional procedure. The study-exclusive sampling for the Ag-RDT PanBio was an NP swab. If NP swabbing was contraindicated for clinical reasons an OP swab was performed. The study-exclusive swab was taken in the same nostril as the routine swab. Laboratory personnel working in both the Ag-RDT testing team and the RT-PCR laboratory were blinded to the results of the other test at all times.

**Antigen-detecting testing.**   Ag-RDT testing was performed in immediate proximity to the sampling in a separate room/container. The Ag-RDT test was started directly after sample collection. The test was conducted as indicated in the IFU, interpreting the test with the naked eye after 15 minutes by two readers blinded to the results of the other. In the case of discrepant results both readers re-interpreted the results and agreed on a final result. Invalid test results were repeated once with the remaining buffer solution in the test tubes.

**RT-PCR testing.**   The collected swabs (Heidelberg, IMPROSWAB, Improve; Berlin, eSwab, Copan) for routine RT-PCR testing were kept in provided Amies solution. The RT-PCR assays performed were Allplex *SARS-CoV-2* Assay from Seegene (Seoul, South Korea) in Heidelberg and the Roche cobas *SARS CoV-2* assay (Pleasanton, CA United States) on the cobas® 6800 or 8800 system or the *SARS CoV-2* assay from TibMolbiol (Berlin,

Germany) in Berlin. Ct-values varied in a range of 2–3 between the three technologies. For the Ct-value presentation and the viral load calculations the E-Gene was used as reference Ct-value. A conversion of the Ct-values into viral-load was performed using quantified specific in vitro-transcribed RNA [14].

**Additional data collection.** All participants were asked to provide additional information about their comorbidities, symptoms, symptom duration and severity of disease (questionnaire available in the, Section (B) in S1 File).

## Data management

All data were collected and managed using Research Electronic Data Capture (REDCap) tools hosted at Heidelberg University [15].

## System usability scale and ease-of-use assessment

A standardized System Usability Scale (SUS) questionnaire and an ease-of-use assessment (EoU) were designed to understand the usability and feasibility of the test [16]. The questionnaire and the EoU survey can be found in the Section (C) and (D) in S1 File. Laboratory personnel from both study sites were invited to complete the questionnaire. An over-all SUS score above 68 is interpreted as above average and anything below the score of 68 is below average [16]. A heat-map was generated to analyse aspects related to the ease-of-use of the test, categorising each as satisfactory, average or dissatisfactory (Section (E) in S1 File and Fig 1). The matrix used for this analysis is also found in the Section (F) in S1 File.

## Statistical analysis

The sensitivity and specificity of the Ag-RDT with 95% confidence intervals (95% CIs) were assessed as per Altman compared to RT-PCR as reference standard (sample size calculations are provided in the S1 File; a statistical analysis plan is available upon request) [17]. Sub-analyses were performed by sampling strategy, symptoms, duration of symptoms, Ct-values and study sites. The significance threshold was set at a two-sided alpha value of 0.05. Participants with an invalid PCR result were excluded from the analysis. All analyses and plots were performed using the R version 4.0.3.

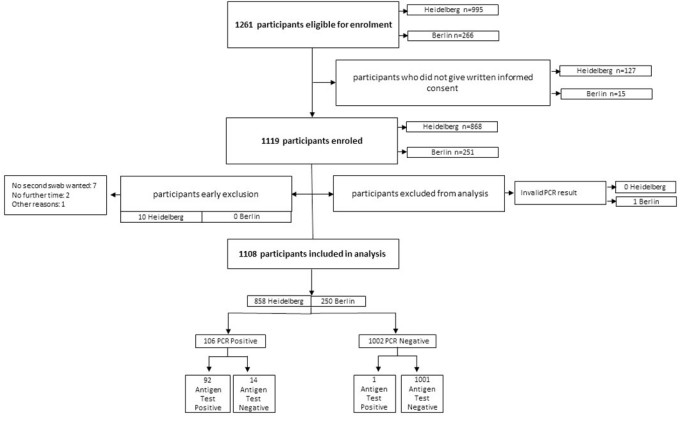

**Fig 1. Study flow.**

## Results

### Clinical diagnostic accuracy

During the enrolment period, from Sept 28, 2020 to Oct 30, 2020, a total of 1261 eligible participants meeting the inclusion criteria were screened for this study. From these 1261 participants, 1119 agreed to undergo a second swab for study purposes only (Fig 1). 10 participants had to be excluded from the study, initially agreeing on participation but denying a second sample collection after the routine swab was performed. After the data cleaning and the exclusion of one invalid PCR test result (N = 1), a total of 1108 participants were included in the analysis. The site in Heidelberg enrolled 858 participants between the Sept 28 and Oct 30, 2020, and the site in Berlin enrolled 250 participants between Oct 19–30, 2020.

The clinical and demographic characteristics of the enrolled participants are summarized in Table 1. The mean age of all participants was 39.4 years (Standard Deviation (SD) 14.1) with Berlin presenting a younger study population compared to Heidelberg. 50.7% of participants were female and 33.4% had comorbidities. 712 participants (64.7%) reported having symptoms on the testing day, with an average symptom duration in days of 4.01 days (SD 3.1). The populations enrolled in Berlin and Heidelberg were significantly different in that participants in Berlin typically enrolled with symptoms (96.8%) while in Heidelberg almost half of the participants were tested based on high-risk contacts without symptoms and compared to the other half of participants reporting symptoms on the testing day (54.8%). Also, participants in Heidelberg were more likely to present earlier in their course of disease (mean 3.5 versus 4.98 days in Berlin) and were more likely to have comorbidities (36.6% in Heidelberg versus 21.2% in Berlin). In total 106 (9.6%) participants were diagnosed with a *SARS-CoV-2* infection by RT-PCR testing during the enrolment period with 23.6% in Berlin and 5.5% in Heidelberg. The mean viral load was 7.4 for both sites with only a slight difference in the SD (Table 1).

The PanBio had an overall sensitivity of 86.8% (92/106 RT-PCR positives detected; 95% Confidence Interval (CI): 79.0% - 92.0%) and a specificity of 99.9% (1 false positive; 95% CI: 99.4%-100%). In a predefined sub analysis by Ct-value, sensitivity for samples that had a Ct-Value $>= 25$ was 66.7% (95% CI: 49.6%-80.2%) and sensitivity for samples with a Ct-value $<25$ was 95.8% (95% CI: 88.5%-98.6%). When samples with a Ct-value $>= 30$ were assessed, the sensitivity was only 33% (95% CI 13.8%-60.1%) but 93.5% (95% CI: 86.6%-97.0%) for

**Table 1. Study population characteristics.**

| | | Overall Total N = 1108 | Heidelberg Total N = 858 | Berlin Total N = 250 |
|---|---|---|---|---|
| **Age in years** Information available on N = 1108 | Mean (SD) | 39.4 (14.1) | 40.3 (14.6) | 34.7 (11.5) |
| **Gender** Information available on N = 1099 | Female N (%) | 557 (50.7) | 455 (53.0) | 102 (40.8) |
| **BMI >25** Information available on N = 1030 | N (%) | 487 (47.3) | 424 (49.4) | 63 (24.8) |
| **Comorbidities** Information available on N = 1100 | N (%) | 367 (33.4) | 314 (36.6) | 53 (21.2) |
| **Symptoms on Testing Day** Information available on N = 1100 | N (%) | 712 (64.7) | 470 (54.8) | 242 (96.8) |
| **Duration of symptoms from day of testing in days** Information available on N = 687 | Mean (SD) | 4.01 (3.1) | 3.50 (2.83) | 4.98 (3.32) |
| **Previous test negative** Information available on N = 885 | Yes N (%) | 250 (28.2) | 217 (25.3) | 33 (13.2) |
| **RT-PCR positives** | N (%) | 106 (9.6) | 47 (5.5) | 59 (23.6) |
| **Viral Load** ($\log_{10}$ SARS-CoV2 RNA copies/ml) Information available on N = 105* | Mean (SD) | 7.4 (1.4) | 7.4 (1.3) | 7.4 (1.5) |

• One RT-PCR positive with a mutation in the E gene was excluded

**Table 2. Subgroup analyses for PanBio.**

| | Overall N (%) | Ag-Test positive/ PCR positive N (%) | Ag-Test negative/ PCR positive N (%) | Ag-Test positive/ PCR negative N (%) | Ag-Test negative/ PCR negative N (%) | Sensitivity % (95% CI) | Specificity % (95% CI) |
|---|---|---|---|---|---|---|---|
| **Sensitivity** | | | | | | | |
| **Overall** | 1108 (100) | 92 (8.3) | 14 (1.3) | 1 (0.1) | 1001 (90.3) | 86.8 (79.0–92.0) | 99.9 (99.4–100) |
| **Heidelberg** | 858 (77.4) | 44 (5.1) | 3 (0.3) | 0 | 811 (94.5) | 93.6 (82.8–97.8) | 100 (99.5–100) |
| **Berlin** | 250 (22.6) | 48 (19.2) | 11 (4.4) | 1 (0.4) | 190 (76.0) | 81.4 (69.6–89.3) | 99.5 (97.1–100) |
| **Sampling strategy—Information available for N = 1108** | | | | | | | |
| **NP swab** | 1034 (93.3) | 91 (8.8) | 13 (1.3) | 1 (0.1) | 929 (89.8) | 87.5 (79.8–92.5) | 99.9 (99.4–100) |
| **OP swab** | 74 (6.7) | 1 (1.4) | 1 (1.4) | 0 | 72 (97.3) | 50.0 (25.6–97.4) | 100 (94.9–100) |
| **Symptom duration—Information available for N = 687** | | | | | | | |
| **0–7 days** Overall | 610 (88.8) | 69 (11.3) | 7 (1.1) | 1 (0.2) | 533 (87.4) | 90.8 (82.2–95.5) | 99.6 (98.9–100) |
| **8–14 days** Overall | 70 (10.2) | 8 (11.4) | 5 (7.1) | 0 | 57 (81.4) | 61.5 (35.5–82.3) | 100 (93.7–100) |
| **Symptomatic versus Asymptomatic—Information available for N = 1100** | | | | | | | |
| **Symptomatic** | 712 (64.7) | 79 (11.1) | 12 (1.7) | 1 (0.1) | 620 (87.1) | 86.8 (78.2–92.3) | 99.8 (99.1–100) |
| **Asymptomatic** | 388 (35.3) | 12 (3.1) | 2 (0.5) | 0 | 374 (96.4) | 85.7 (60.1–96·0) | 100 (99.0–100) |
| **Ct-Value PCR <30 and > = 30—Information available for N = 105\*** | | | | | | | |
| **CT value PCR <30** | 93 (88) | 87 (93.5) | 6 (6.5) | NA | NA | 93.5 (86.6–97.0) | NA |
| **CT value PCR > = 30** | 12 (11.0) | 4 (33.0) | 8 (67.0) | NA | NA | 33 (13.8–60.1) | NA |
| **Ct-Value PCR <25 and > = 25—Information available for N = 105\*** | | | | | | | |
| **CT value PCR <25** | 72 (68.0) | 69 (95.8) | 3 (4.2) | NA | NA | 95.8 (88.5–98.6) | NA |
| **CT value PCR > = 25** | 33 (31.0) | 22 (66.7) | 11 (33.3) | NA | NA | 66.7 (49.6–80.2) | NA |

• One RT-PCR positive with a mutation in the E gene was excluded

samples with a Ct-value <30 (see Table 2 and Fig 2). The sensitivity decreased as viral load decreased (Fig 2). A detailed table summarizing the symptoms for Ag-RDT positive and negative participants, the viral load equivalents to the Ct-values and the RT-PCR reference standard is provided in the Section (G) and (H) in S1 File.

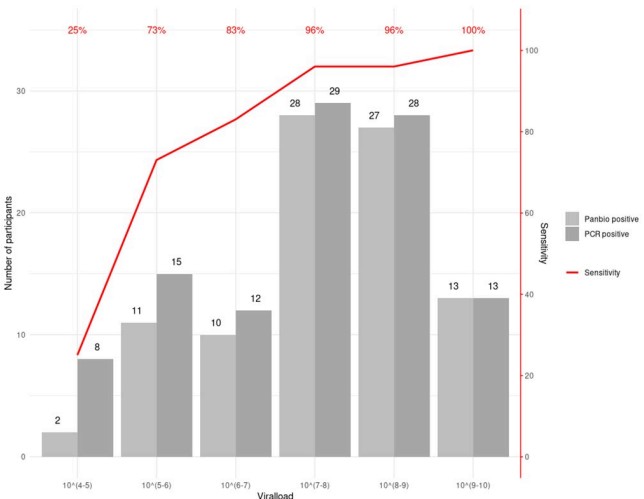

**Fig 2. Sensitivity of PanBio Ag-RDT compared to viral load for all PCR positive cases (105 participants).**

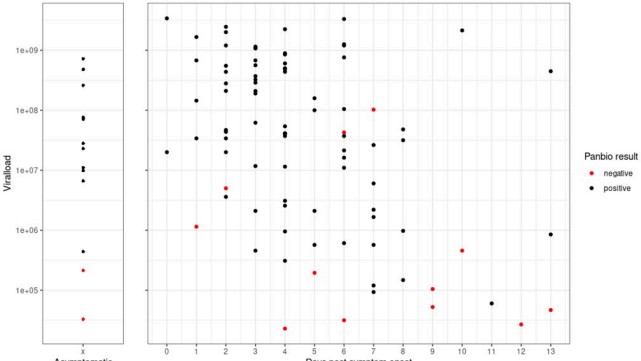

**Fig 3. Viral load and Ag-RDT results for asymptomatic participants and by days post symptom onset for all PCR positive cases (105 participants).**

When assessing test performance by duration of symptoms, we found PanBio performed well in the first 7 days after symptom onset (sensitivity 90.8% (95% CI: 82.2%-95.5%)), with declining sensitivity thereafter (>7 days of symptoms, sensitivity 61.5% (95% CI: 35.5%-82.3%)). This decrease in sensitivity with prolonged symptom duration is also shown in Fig 3, presenting the performance of PanBio with increasing days since symptom onset in relation to the calculated viral load for symptomatic and asymptomatic participants and the Ag-RDT performance.

Out of the total 106 positive RT-PCR cases, 14 participants were without symptoms but with recent high-risk contacts. Within this small participant group the sensitivity of the Ag-RDT was 85.7% (60.1%-96.0%), which compares to the sensitivity of symptomatic participants at 86.8% (78.2%-92.3%); Table 2). Mean Ct-value in asymptomatic was 22.1 (SD 4.4) versus 23.1 (SD 5.0) in symptomatic participants.

The interrater reliability with kappa of 0.99 suggests that the tests results are clearly interpretable. PanBio scored 86 out of 100 points in the SUS showing a test which is easy to use. Problems were encountered when applying the exact amount of the required five drops to the test device (Section (E) in S1 File).

## Discussion

This prospective multi-centre clinical diagnostic accuracy study shows that the PanBio Ag-RDT from Abbott has a good sensitivity of 86.8% and a very high specificity of 99.9% for symptomatic participants and participants with confirmed recent high-risk contacts compared to the reference standard RT-PCR. The Ag-RDT is easily performed in a point-of-care setting.

The differences observed in sensitivity between the two enrolment sites are probably explained by the different stages of the pandemic control. Berlin had a substantially higher prevalence with primarily symptomatic patients being tested and testing occurring later in the disease. Although the mean viral load was the same, the distribution of the viral load was not the same and more participants presented with low viral loads in Berlin, later in the course of infection. There were 15 patients presenting with viral loads <6 log10 copies/mL versus only 8 in Heidelberg. The samples with low viral loads, were responsible for 8 out of the 11 discordant results (false-negative) in Berlin, and 2 out of 3 in Heidelberg. Viral load dynamics play an important role in disease transmission, often rising before symptom onset with an observed peak at the time of symptom onset or at day 3–5 of the disease and a rapid decrease within the following days [18].

With a performance of 93.5% for CT value <30 and 90.8% within the first 7 days of symptoms, the test is likely to detect the vast majority of people with transmission potential *SARS-CoV2* infections, supporting recent published literature [10, 19, 20]. These findings show that the PanBio Ag-RDT has a great potential in a public health setting, identifying the transmission relevant infections within the first 7 days of symptoms. Within the limitation of what can be concluded due to the small sample size of participants without symptoms at the time of testing and the fact that these participants were asymptomatic high-risk contacts presenting early in the disease, the performance of the test was as good as in symptomatic patients with a sensitivity respectively of 85.7% versus 86.8%. This suggests the PanBio to be an option for screening independent of symptoms, especially when the time of exposure is known and is supported by recent data suggesting that viral load in adults does not differ between asymptomatic and symptomatic infections [21].

Considering the test's ease-of-use and the rapid turn-around time between 15 and 20 minutes, along with its high specificity, it could be considered for several use-cases: (1) screening in advance of events at high-risk of transmission (e.g. aggregated settings where contact cannot be avoided); or (2) necessary encounters with persons at high-risk for severe disease of *SARS-CoV-2* (e.g. visitor in nursing homes) in addition to (3) the use in symptomatic patients within the first week of illness when RT-PCR is not available or together with RT-PCR, when a rapid decision is necessary.

Furthermore, given that supervised self-sampling from the anterior nose is a reliable alternative to professional nasopharyngeal sampling, scale-up of testing appears possible without requiring large numbers of trained health-care workers [22, 23].

Overall, our study has several strengths. The population enrolled for testing was representative of the pandemic observed in adults in Germany with a broad spectrum of clinical presentations (from asymptomatic with high-risk contacts to severely ill). Due to the wide-spread testing available and the good test and trace capabilities, the population tested is expected to be a representative spectrum of disease. Also, the tests were performed at POC thus mimicking the real-world challenges of POC testing. Lastly, the comprehensive ease-of-use assessment highlighted important points for operationalization of the test.

However, the study also has several limitations. First, it was conducted only in one country, thus making it less representative of the pandemic at large. Second, the reference standard testing was performed on an NP swab in Heidelberg versus an NP/OP swab in Berlin. However, a recent systematic review does not suggest those sampling methods to yield different results [24]. Thirdly, participants without symptoms at the time of testing could potentially have developed symptoms after being tested, yet this information was not recorded during this study. Furthermore, the time of exposure to high-risk contacts for participants without symptoms was not recorded, giving the possibility that these participants presented very early in the course of disease, maybe resulting in false-negatives [25]. And lastly, we performed different PCR methods as a reference standard recognizing the limitations of this method However, we aimed to provide comparability between methods by calibrating the methods and in addition reporting on viral load. Furthermore, we acknowledge the limitation of the PCR method as a reference standard, as it is not always a meaningful test when considering viable virus and risk of transmission [26]. Thus, using the PCR reference standard, we might have underestimated the performance of the Ag-RDT when it comes to detection of viable virus.

In summary, the favourable ease-of-use results and the limited infrastructure required for the Ag-RDT testing procedure, its high specificity in addition to the high sensitivity of the test in persons with high viral load, can empower control of population transmission if implemented in well-designed testing programs [27–29]. Policy makers should move from

considering only test sensitivity to more holistic testing strategies, incorporating Ag-RDTs in addition to and in combination with RT-PCR to optimize the reach and depth of testing.

## Supporting information

**S1 File.**
(DOCX)

## Acknowledgments

We acknowledge the work of Angelika Sandritter in supporting the organisation and employees of this study. In addition, we thank the team of the KTS and the local Health Department Rhein-Neckar for their support on the testing site.

## Author Contributions

**Conceptualization:** Lisa J. Krüger, Mary Gaeddert, Claudius Gottschalk, Julian A. F. Klein, Andreas K. Lindner, Nira R. Pollock, Britta Knorr, Andreas Welker, Jilian A. Sacks, Claudia M. Denkinger.

**Data curation:** Mary Gaeddert, Frank Tobian.

**Formal analysis:** Frank Tobian, Margaretha de Vos, Claudia M. Denkinger.

**Investigation:** Lisa J. Krüger, Mary Gaeddert, Federica Lainati, Julian A. F. Klein, Olga Nikolai, Andreas K. Lindner, Frank P. Mockenhaupt, Joachim Seybold, Victor M. Corman, Christian Drosten, Margaretha de Vos, Claudia M. Denkinger.

**Methodology:** Lisa J. Krüger, Andreas K. Lindner, Victor M. Corman, Nira R. Pollock, Jilian A. Sacks, Claudia M. Denkinger.

**Project administration:** Lisa J. Krüger, Julian A. F. Klein, Paul Schnitzler, Olga Nikolai, Andreas K. Lindner, Christian Drosten, Andreas Welker, Claudia M. Denkinger.

**Resources:** Hans-Georg Kräusslich, Frank P. Mockenhaupt, Joachim Seybold, Victor M. Corman, Christian Drosten, Britta Knorr, Andreas Welker, Jilian A. Sacks, Claudia M. Denkinger.

**Software:** Mary Gaeddert, Frank Tobian, Federica Lainati, Paul Schnitzler, Victor M. Corman, Christian Drosten, Andreas Welker, Margaretha de Vos, Claudia M. Denkinger.

**Supervision:** Lisa J. Krüger, Mary Gaeddert, Julian A. F. Klein, Hans-Georg Kräusslich, Olga Nikolai, Andreas K. Lindner, Frank P. Mockenhaupt, Joachim Seybold, Victor M. Corman, Christian Drosten, Britta Knorr, Andreas Welker, Jilian A. Sacks, Claudia M. Denkinger.

**Validation:** Federica Lainati, Claudius Gottschalk, Paul Schnitzler.

**Visualization:** Andreas K. Lindner, Nira R. Pollock, Andreas Welker, Claudia M. Denkinger.

**Writing – original draft:** Lisa J. Krüger, Nira R. Pollock, Jilian A. Sacks, Claudia M. Denkinger.

**Writing – review & editing:** Lisa J. Krüger, Mary Gaeddert, Frank Tobian, Federica Lainati, Claudius Gottschalk, Julian A. F. Klein, Paul Schnitzler, Olga Nikolai, Andreas K. Lindner, Frank P. Mockenhaupt, Joachim Seybold, Nira R. Pollock, Britta Knorr, Andreas Welker, Margaretha de Vos, Jilian A. Sacks, Claudia M. Denkinger.

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
