## [Decision Letter · Decision Letter 0]

7 Jan 2021

PONE-D-20-40313

The Abbott PanBio WHO emergency use listed, rapid, antigen-detecting point-of-care diagnostic test for SARS-CoV-2

-

Evaluation of the accuracy and ease-of-use

PLOS ONE

Dear Dr. Denkinger,

Thank you for submitting your manuscript to PLOS ONE. After careful consideration, we feel that it has merit but does not fully meet PLOS ONE’s publication criteria as it currently stands. Therefore, we invite you to submit a revised version of the manuscript that addresses the points raised during the review process.

Both Reviewers were very enthusiastic about the manuscript  and have provided lists of minor corrections/edits to be made. Please note that Reviewer 2's has several more substantive comments that should be addressed.

We look forward to receiving your revised manuscript.

Kind regards,

Nicholas J Mantis

Academic Editor

PLOS ONE

Journal Requirements:

2. Please provide a sample size and power calculation in the Methods, or discuss the reasons for not performing one before study initiation.

Reviewers' comments:

Reviewer's Responses to Questions

**Comments to the Author**

1. Is the manuscript technically sound, and do the data support the conclusions?

Reviewer #1: Yes

Reviewer #2: Partly

2. Has the statistical analysis been performed appropriately and rigorously? 

Reviewer #1: Yes

Reviewer #2: Yes

3. Have the authors made all data underlying the findings in their manuscript fully available?

Reviewer #1: Yes

Reviewer #2: Yes

4. Is the manuscript presented in an intelligible fashion and written in standard English?

Reviewer #1: Yes

Reviewer #2: Yes

5. Review Comments to the Author

Reviewer #1: This is an excellent manuscript and I agree is an important contribution to the field about an important topic that should be accepted for publication after a minor revision.

I would change "CT" to "Ct" for cycle threshold as this is the convention

Line 39: would change "days of symptoms" to "day after symptom onset"

Line 44: change "days from symptom onset" to "days after symptom onset"

Line 51-52: sentence is awkward. Perhaps change to: "In March 2020, the World Health Organisation (WHO) emphasized the importance of access to testing..."

Line 53: would delete "all" from "remains the gold standard among all diagnostic tests"

Line 55: Would add a point about the limitations of PCR testing given the prolonged PCR positivity (not always a meaningful test in terms of telling whether someone has active replicating virus or transmission potential, see: https://wwwnc.cdc.gov/eid/article/26/8/20-1097_article

Line 64-65: sentence is awkward. Consider changing to "This manufacturer-independent study follows a WHO-approved..."

Line 69: would change "an excellent" to "a" to make more neutral

Line 72: I believe this is your first use of "POC" would add "point of care (POC)" since the first time it is being used

Line 87-88: This is not clear. Perhaps "Colloidal gold conjugated antibodies on the membrane strip react with viral antigens generating a colour change in the device window..."

Line 89: Awkward sentence. Perhaps change to "The results must be interpreting following 15 to 20 minutes of incubation..."

Line 98: change "in English or German" to "of English or German"

Line 98: change "a protocol" to "The study protocol"

Line 192: Would be helpful to know day of exposure relative to testing for all asymptomatic cases. If not known would write "14 participants without symptoms with recent high-risk contacts."

Line 203: would note that this sensitivity and specificity is for people with symptoms or a recent close confirmed contact with SARS-CoV-2 infection

Would add a new paragraph between your current first and second paragraphs in the discussion quickly reviewing viral load dynamics. Ie viral loads rise before symptom onset and peak around symptom onset and decline quickly after symptom onset. Should also note that the period of transmission for immunocompetent adults with symptomatic infection is from a couple days before symptom onset to around a week after symptom onset. Here are a couple reviews of the topic: https://wwwnc.cdc.gov/eid/article/26/8/20-1097_article, https://www.thelancet.com/journals/lanmic/article/PIIS2666-5247(20)30172-5/fulltext

Line 206: change "overall viral load" to "mean viral load"

Line 208: would change to "low viral loads in Berlin, later in the course of infection."

Line 210: would delete "false-negative" and change to "discordant results"

Line 212: would change perhaps to "is likely to detect the vast majority of people with SARS-CoV-2 infections with transmission potential,..."

Line 213: Add a sentence reminding readers that since sensitivity is so good during first week of symptoms (which is the period of transmission) these tests have great utility as public health tools.

Line 214: change "asymptomatic participants" to "participants without symptoms at the time of testing"

Line 216: would add a sentence noting that you did not assess whether these individuals eventually develop symptoms and could have therefore been presymptomatic at the time of testing. Given their high viral loads they are presumed to have transmission potential so very important for them to get a positive result so they can isolate.

Line 224: change to "the use in symptomatic patients within the first week of illness"

Line 241: Would add that a limitation is that for asymptomatic cases the time of exposure to high risk cases was not recorded. Note that if tests are done to early they cannot exclude infection since viral load may not yet have had time to rise. The importance of this is that a negative test performed too early should not be used to support removing an individual from quarantine if a high risk exposure was very recent. Would cite: https://www.acpjournals.org/doi/10.7326/M20-1495

Reviewer #2: The authors evaluated the performance of the Panbio COVID-19 Ag Rapid Test Device for SARS-CoV-2 in nasopharyngeal swab specimens. For the ease of communication, Panbio stands for this kit in this review.

It was a prospective multicenter study, conducted in two sites in Germany, one was in Heidelberg while another was in Berlin. A total of 1108 individuals were recruited. Paired swabs were obtained for each individual, the ‘routine swab’ was used for RT-PCR while the ‘second study-exclusive swab’ was used for Panbio test.

Among these 1108 individuals, 106 were positive by RT-PCR. Among these 106 RT-PCR positive, 92 were Panbio positive, the sensitivity of the Panbio test was 86.8% (92/106).

There are two comments about this manuscript:

Comment 1

Lines 102-107

The description of the swabs used for RT-PCR and Panbio was confusing. There are four different terms of swabs: (1) routine swab, (2) nasopharyngeal swab, (3) combined nasopharyngeal and oropharyngeal swab, (4) study-exclusive swab.

Comment 2

There are at least five peer review articles evaluating the Panbio kit:

1. Linares et al. Panbio antigen rapid test is reliable to diagnose SARS-CoV-2 infection in the first 7 days after the onset of symptoms. J Clin Virol. 2020 Oct 16;133:104659. doi: 10.1016/j.jcv.2020.104659.

2. Fenollar et al. Evaluation of the Panbio Covid-19 rapid antigen detection test device for the screening of patients with Covid-19. J Clin Microbiol. 2020 Nov 2:JCM.02589-20. doi: 10.1128/JCM.02589-20.

3. Lanser et al. Evaluating the clinical utility and sensitivity of SARS-CoV-2 antigen testing in relation to RT-PCR Ct values. Infection. 2020 Nov 13. doi: 10.1007/s15010-020-01542-0.

4. Albert et al. Field evaluation of a rapid antigen test (Panbio™ COVID-19 Ag Rapid Test Device) for COVID-19 diagnosis in primary healthcare centers. Clin Microbiol Infect. 2020 Nov 12:S1198-743X(20)30697-2. doi: 10.1016/j.cmi.2020.11.004.

Based on these four articles, the sensitivity of the Panbio ranged from 60.8% to 79.6%.

5. Gremmels, H., et al., Real-life validation of the Panbio COVID-19 Antigen Rapid Test (Abbott) in community-dwelling subjects with symptoms of potential SARS-CoV-2 infection. EClinicalMedicine Available online 5 December 2020 https://doi.org/10.1016/j.eclinm.2020.100677

Table 2.

The authors reported high sensitivity of Panbio test 86.8% (92/106). It might be due to the high viral load study groups enrolled:

Figrue 2.

70 individuals (66.7%) with viral load >= 10^(7-8)

35 individuals (33.3%) with viral load <= 10^(6-7)

Lines 204-210

The authors also discussed the sensitivity difference observed between Berlin and Heidelberg.

The authors have to state the reasons for the high sensitivity of Panbio test clearly when comparing with those five studies (listed above), sensitivity ranged from 60.8%-81.1%.

6. PLOS authors have the option to publish the peer review history of their article (what does this mean?). If published, this will include your full peer review and any attached files.

Reviewer #1: No

Reviewer #2: No

---

## [Author Response · Author response to Decision Letter 0]

10 Feb 2021

Response to Reviewers

Journal Comments to the Author Response from Author

1. If there are ethical or legal restrictions on sharing a de-identified data set, please explain them in detail (e.g., data contain potentially identifying or sensitive patient information) and who has imposed them (e.g., an ethics committee). Please also provide contact information for a data access committee, ethics committee, or other institutional body to which data requests may be sent. 

Every participant signed a written consent form including a data protection section. Within this section it is clearly stated that the data will be pseudo anonymized for the data analysis. In addition, it is stated that the data will only be used for purposes of this study. The ethical committee of the University Heidelberg reviewed the study protocol and approved it on the 23.03.2020. 

2. If there are no restrictions, please upload the minimal anonymized data set necessary to replicate your study findings as either Supporting Information files or to a stable, public repository and provide us with the relevant URLs, DOIs, or accession numbers. Please see http://www.bmj.com/content/340/bmj.c181.long for guidelines on how to de-identify and prepare clinical data for publication. For a list of acceptable repositories, please see 

http://journals.plos.org/plosone/s/data-availability#loc-recommended-repositories. 

A data set, including the anonymized data will be uploaded to a data repository and the link will be shared. 

Please provide a sample size and power calculation in the Methods, or discuss the reasons for not performing one before study initiation. The sample size calculations are included in the supplement.

Reviewer 1 Comments to the Author Response from Author

1. I would change "CT" to "Ct" for cycle threshold as this is the convention 

Thanks, for this helpful comment, we have included the changes throughout the whole paper. 

2. Line 39: would change "days of symptoms" to "day after symptom onset" and 

Line 44: change "days from symptom onset" to "days after symptom onset" 

Thank you for this comment, the changes have been included in the manuscript. 

3. Line 51-52: sentence is awkward. Perhaps change to: "In March 2020, the World Health Organization (WHO) emphasized the importance of access to testing..." 

Sentence has been changed and hopefully made more understandable, thank you for this remark. 

4. Line 53: would delete "all" from "remains the gold standard among all diagnostic tests" We appreciate this suggestion and have included it in the manuscript. 

5. Line 55: Would add a point about the limitations of PCR testing given the prolonged PCR positivity (not always a meaningful test in terms of telling whether someone has active replicating virus or transmission potential, see: https://wwwnc.cdc.gov/eid/article/26/8/20-1097_article We appreciate this comment and do understand that the RT-PCR method as a gold standard has its limitations. Yet, we would like to focus on the RT-PCR as the currently accepted gold standard as a comparison for the evaluation of PanBio. We have added text on the limitations of the PCR in the discussion

6. Line 64-65: sentence is awkward. Consider changing to "This manufacturer-independent study follows a WHO-approved..." We appreciate this suggestion and included it as recommended in the manuscript. 

7. Line 69: would change "an excellent" to "a" to make more neutral We appreciate this recommendation and changed the wording to a more neutral way. 

8. Line 72: I believe this is your first use of "POC" would add "point of care (POC)" since the first time it is being used We appreciate this suggestion and included it as recommended in the manuscript.

9. Line 87-88: This is not clear. Perhaps "Colloidal gold conjugated antibodies on the membrane strip react with viral antigens generating a colour change in the device window..." We appreciate this suggestion and included it as recommended in the manuscript.

10. Line 89: Awkward sentence. Perhaps change to "The results must be interpreting following 15 to 20 minutes of incubation..." Thank you. We appreciate this suggestion and included it as recommended in the manuscript.

11. Line 98: change "in English or German" to "of English or German" We appreciate this suggestion and included it as recommended in the manuscript.

12. Line 98: change "a protocol" to "The study protocol" We appreciate this suggestion and included it as recommended in the manuscript.

13. Line 192: Would be helpful to know day of exposure relative to testing for all asymptomatic cases. If not known would write "14 participants without symptoms with recent high-risk contacts." We do not have the exact data on day of exposure and therefore we appreciate this suggestion and changed the manuscript accordingly. 

14. Line 203: would note that this sensitivity and specificity is for people with symptoms or a recent close confirmed contact with SARS-CoV-2 infection We appreciate this suggestion and included it as recommended in the manuscript.

15. Would add a new paragraph between your current first and second paragraphs in the discussion quickly reviewing viral load dynamics. Ie viral loads rise before symptom onset and peak around symptom onset and decline quickly after symptom onset. Should also note that the period of transmission for immunocompetent adults with symptomatic infection is from a couple days before symptom onset to around a week after symptom onset. Here are a couple reviews of the topic: https://wwwnc.cdc.gov/eid/article/26/8/20-1097_article, https://www.thelancet.com/journals/lanmic/article/PIIS2666-5247(20)30172-5/fulltext Thank you for this recommendation. A short explanation has been added at line 259 to 261. 

16. Line 206: change "overall viral load" to "mean viral load" We appreciate this suggestion and included it as recommended in the manuscript.

17. Line 208: would change to "low viral loads in Berlin, later in the course of infection." We appreciate this suggestion and included it as recommended in the manuscript.

18. Line 210: would delete "false-negative" and change to "discordant results" We appreciate this suggestion and have added discordant results to the sentences leaving false-negative in brackets for the clarification of these discordant results. 

19. Line 212: would change perhaps to "is likely to detect the vast majority of people with SARS-CoV-2 infections with transmission potential,..." We appreciate this suggestion and included it as recommended in the manuscript.

20. Line 213: Add a sentence reminding readers that since sensitivity is so good during first week of symptoms (which is the period of transmission) these tests have great utility as public health tools. Thank you for this recommendation, a sentence has been added to the manuscript. 

21. Line 214: change "asymptomatic participants" to "participants without symptoms at the time of testing" We appreciate this suggestion and included it as recommended in the manuscript.

22. Line 216: would add a sentence noting that you did not assess whether these individuals eventually develop symptoms and could have therefore been presymptomatic at the time of testing. Given their high viral loads they are presumed to have transmission potential so very important for them to get a positive result so they can isolate. Thank you for this recommendation, we have added a sentence to the manuscript explaining that data on later developed symptoms was not recorded during out study. 

23. Line 224: change to "the use in symptomatic patients within the first week of illness" We appreciate this suggestion and included it as recommended in the manuscript.

24. Line 241: Would add that a limitation is that for asymptomatic cases the time of exposure to high-risk cases was not recorded. Note that if tests are done to early, they cannot exclude infection since viral load may not yet have had time to rise. The importance of this is that a negative test performed too early should not be used to support removing an individual from quarantine if a high-risk exposure was very recent. Would cite: https://www.acpjournals.org/doi/10.7326/M20-1495 Thank you for this recommendation, we have not recorded these data and have added this as a clear limitation to the study. 

Reviewer 2 Comments to the Author Response from Author

1. Lines 102-107

The description of the swabs used for RT-PCR and Panbio was confusing. There are four different terms of swabs: (1) routine swab, (2) nasopharyngeal swab, (3) combined nasopharyngeal and oropharyngeal swab, (4) study-exclusive swab. We appreciate this comment, thank you very much. We have added words to clarify the explanation for the reader and hope this makes this paragraph less confusing. 

2. There are at least five peer review articles evaluating the Panbio kit:

1. Linares et al. Panbio antigen rapid test is reliable to diagnose SARS-CoV-2 infection in the first 7 days after the onset of symptoms. J Clin Virol. 2020 Oct 16;133:104659. doi: 10.1016/j.jcv.2020.104659.

2. Fenollar et al. Evaluation of the Panbio Covid-19 rapid antigen detection test device for the screening of patients with Covid-19. J Clin Microbiol. 2020 Nov 2:JCM.02589-20. doi: 10.1128/JCM.02589-20.

3. Lanser et al. Evaluating the clinical utility and sensitivity of SARS-CoV-2 antigen testing in relation to RT-PCR Ct values. Infection. 2020 Nov 13. doi: 10.1007/s15010-020-01542-0.

4. Albert et al. Field evaluation of a rapid antigen test (Panbio™ COVID-19 Ag Rapid Test Device) for COVID-19 diagnosis in primary healthcare centers. Clin Microbiol Infect. 2020 Nov 12:S1198-743X(20)30697-2. doi: 10.1016/j.cmi.2020.11.004.

Based on these four articles, the sensitivity of the Panbio ranged from 60.8% to 79.6%.

5. Gremmels, H., et al., Real-life validation of the Panbio COVID-19 Antigen Rapid Test (Abbott) in community-dwelling subjects with symptoms of potential SARS-CoV-2 infection. EClinicalMedicine Available online 5 December 2020 https://doi.org/10.1016/j.eclinm.2020.100677 Thank you for this comment, we have reviewed the additionally mentioned publications and have included these in the manuscript, starting in the introduction and the discussion. 

3. Table 2.

The authors reported high sensitivity of Panbio test 86.8% (92/106). It might be due to the high viral load study groups enrolled:

Figure 2.

70 individuals (66.7%) with viral load >= 10^(7-8)

35 individuals (33.3%) with viral load <= 10^(6-7) We appreciate this comment, thank you. In the discussion we have added a paragraph on viral load dynamics and pointed out that the test was very sensitive in the first week of symptoms, understood to be the week with the highest viral loads.

4. Lines 204-210

The authors also discussed the sensitivity difference observed between Berlin and Heidelberg.

The authors have to state the reasons for the high sensitivity of Panbio test clearly when comparing with those five studies (listed above), sensitivity ranged from 60.8%-81.1%. Thank you for this comment. We have included this information in the discussion and it can be observed that most studies support the high sensitivity within the first week of illness.

---

## [Editor Report · Decision Letter 1]

17 Feb 2021

The Abbott PanBio WHO emergency use listed, rapid, antigen-detecting point-of-care diagnostic test for SARS-CoV-2

-

Evaluation of the accuracy and ease-of-use

PONE-D-20-40313R1

Dear Dr. Denkinger,

We’re pleased to inform you that your manuscript has been judged scientifically suitable for publication and will be formally accepted for publication once it meets all outstanding technical requirements.

Kind regards,

Nicholas J Mantis

Academic Editor

PLOS ONE
---

## [Editor Report · Acceptance letter]

3 May 2021

PONE-D-20-40313R1 

The Abbott PanBio WHO emergency use listed, rapid, antigen-detecting point-of-care diagnostic test for *SARS-CoV-2*
-
Evaluation of the accuracy and ease-of-use 

Dear Dr. Denkinger:

I'm pleased to inform you that your manuscript has been deemed suitable for publication in PLOS ONE. Congratulations! Your manuscript is now with our production department. 

Kind regards, 

on behalf of

Dr. Nicholas J Mantis 

Academic Editor

PLOS ONE